# The Effectiveness of Tobacco Dependence Education in Health Professional Students’ Practice: A Systematic Review and Meta-Analysis of Randomized Controlled Trials

**DOI:** 10.3390/ijerph16214158

**Published:** 2019-10-28

**Authors:** Kathryn Hyndman, Roger E. Thomas, H. Rainer Schira, Jenifer Bradley, Kathryn Chachula, Steven K. Patterson, Sharon M. Compton

**Affiliations:** 1Faculty of Health Studies, Brandon University, Brandon, MB R7A 6A9, Canada; chachulak@brandonu.ca; 2School of Medicine, University of Calgary, Calgary, AB T2N 4N1, Canada; 3John E. Robbins Library, Brandon University, Brandon, MB R7A 6A9, Canada; SchiraR@BrandonU.CA; 4Department of National Defence, Petawawa, ON K8H 2X3, Canada; jeniferbradley@hotmail.ca; 5School of Dentistry, Faculty of Medicine & Dentistry, University of Alberta, Edmonton, AB T6G 1C9, Canada; steven.patterson@ualberta.ca; 6Dental Hygiene Program, Faculty of Medicine & Dentistry, University of Alberta, Edmonton, AB T6G 1C9, Canada; Sharon.compton@ualberta.ca

**Keywords:** health professional students, smoking cessation, tobacco dependence intervention, education, randomized controlled trials, systematic review

## Abstract

The objective of this study was to perform a systematic review to examine the effectiveness of tobacco dependence education versus usual or no tobacco dependence education on entry-level health professional student practice and client smoking cessation. Sixteen published databases, seven grey literature databases/websites, publishers’ websites, books, and pertinent reference lists were searched. Studies from 16 health professional programs yielded 28 RCTs with data on 4343 healthcare students and 3122 patients. Two researchers independently assessed articles and abstracted data about student knowledge, self-efficacy, performance of tobacco cessation interventions, and patient smoking cessation. All forms of tobacco were included. We did not find separate interventions for different kinds of tobacco such as pipes or flavoured tobacco. We computed effect sizes using a random-effects model and applied meta-analytic procedures to 13 RCTs that provided data for meta-analysis. Students’ counseling skills increased significantly following the 5As model (SMD = 1.03; 95% CI 0.07, 1.98; *p* < 0.00001, I^2^ 94%; *p* = 0.04) or motivational interviewing approach (SMD = 0.90, 95% CI 0.59, 1.21; *p* = 0.68, I^2^ 0%; *p* < 0.00001). With tobacco dependence counseling, 78 more patients per 1000 (than control) reported quitting at 6 months (OR 2.02; 95% CI 1.49, 2.74, I^2^ = 0%, *p* = 0.76; *p* < 0.00001), although the strength of evidence was moderate or low. Student tobacco cessation counseling improved guided by the above models, active learning strategies, and practice with standardized patients.

## 1. Introduction

Bilano and colleagues project that by 2025 more than one billion people will remain tobacco users if recent trends remain unchanged [1]. Tobacco use may account for over eight million deaths by 2030, with over 80% in low- and middle-income countries [2]. Recent meta-analyses provide evidence that tobacco use is a risk factor for cancer, heart and cerebrovascular disease, chronic obstructive pulmonary disease, diabetes mellitus, nephritis, and chronic liver disease [3,4,5,6,7,8,9,10], chronic kidney disease [11], bladder cancer [12], multiple sclerosis [13], poor surgical outcomes [14,15,16,17,18,19], pregnancy complications [20,21,22], and poor oral health and tooth loss [23,24].

Educating health professionals to address tobacco use and dependence has been endorsed in Canada [25,26] and globally [27,28,29]. Article 12 (a, d) and Article 14 of the World Health Organization (WHO) Framework Convention on Tobacco Control (FCTC) describe the need for comprehensive education programs for health professionals to address health and addiction risks of tobacco consumption, as well as treatment of tobacco use and dependence within a comprehensive tobacco control strategy [29]. Tobacco dependence treatment includes brief advice and counseling for smoking prevention and cessation and recommending or prescribing approved cessation medications [27,29,30]. Health professionals who receive training in smoking cessation counseling are one-and-a-half- to two-times more likely to offer clients smoking cessation interventions and client quit rates increase when counseling is delivered by a variety of healthcare providers [30,31,32,33,34,35,36]. Health professionals continue to report a lack of training and doubt their effectiveness in treating client tobacco use and dependence despite having direct access to clients who smoke [25,31,37,38,39,40,41,42,43,44] and available evidence-based guidelines [30,45,46].

In this systematic review, we assessed entry-level education programs which prepare students for entry into regulated health disciplines and for providing direct patient care. Entry-level education aims to prepare future health professionals for a desired standard of practice where all providers would intervene with tobacco users and offer a brief intervention at each health visit [27,28,29,30,47,48]. The 5As model [30] (Ask, Advise, Assess, Assist, Arrange) has been in use since 1990 in clinical practice guidelines (CPGs) [30,45,46,49] and in educational curricula [28,47,48,50,51,52,53,54,55,56,57,58,59,60,61,62]. This brief intervention approach includes asking all clients if they use tobacco, advising them not to start smoking or to quit tobacco use, assessing clients for interest in quitting if they use tobacco, and assisting them to quit using discipline-specific interventions. Arrange involves scheduling follow-up contact that could be a return visit, a telephone call, or a referral for more intensive treatment. A key question is whether entry-level education equips students to assess and treat client tobacco use and dependence [63,64,65,66,67,68,69,70]. An understanding of the effects of educational programming on health professional student performance in treating client tobacco use and dependence and on clients’ quitting behaviors would enhance health professional schools’ capacity to equip graduates with the knowledge and skills to address an identified global health priority. This is the first systematic review to examine the effect of tobacco dependence education implemented in 16 disciplines on students’ practice in treating client tobacco use and dependence.

### Review Question

What is the effect of entry-level tobacco dependence education on (1) health professional students’ knowledge and self-efficacy, (2) performance of the specific skills in the individual tobacco cessation interventions, and (3) client smoking cessation behaviors?

## 2. Materials and Methods

This systematic review followed an a priori published protocol [71]. The protocol was registered with the international prospective register of systematic reviews (PROSPERO) on 8 August 2016 as CRD42016044106. We utilized the Joanna Briggs Institute (JBI) methodology [72] and the Cochrane Handbook [73] for systematic reviews of effectiveness. The RevMan 5.3 program generated Forest plots [74].

### 2.1. Criteria for Inclusion

#### 2.1.1. Inclusion Criteria: Participants

This review considered studies of health professional students within 16 programs of study and 23 post-graduate medical programs (Table 1). Medical resident trainees and advanced practice nursing students were included as entry-level because they provide specialized care to patients with health concerns commonly linked to tobacco use. We excluded non-entry-level educational programs, programs in which graduates do not have daily, direct interactions with clients, and programs from non-regulated disciplines whose graduates can practice without qualifying for a license.

#### 2.1.2. Inclusion Criteria: Intervention

This review considered reports on the implementation and evaluation of an entry-level program, curricular activity or component in smoking prevention or smoking cessation, and its impact on student practice in promoting client health. The educational intervention and its impact on student practice must have occurred during the entry-level program. The review described the nature of the intervention delivery, settings, and models used.

#### 2.1.3. Inclusion Criteria: Comparator

This review considered studies that compared the educational intervention (including pre- and post-training results) to usual or no educational program. 

#### 2.1.4. Inclusion Criteria: Outcomes

This review considered studies that reported evaluations of health professional student outcomes in treating tobacco use and dependence: knowledge, self-efficacy, and clinical performance of tobacco prevention and cessation interventions. Student knowledge was measured by scores on exams and questionnaires and student efficacy in treating tobacco use and dependence by instruments designed to rate self-confidence in treating tobacco use. Student clinical performance was assessed by faculty observations, standardized patients (SPs), and client reports. The primary client outcome, smoking cessation, was measured by client reports or biochemical measures of smoking abstinence at either six- or 12-months following smoking cessation intervention.

#### 2.1.5. Inclusion Criteria: Study Types

All randomized controlled trials (RCTs) that evaluated the effectiveness of educational programming on student knowledge, self-efficacy and performance of tobacco use prevention and treatment interventions were included. Studies were included that reported pre- and post-training results and compared the findings to a control or comparison group experiencing usual teaching or another educational intervention. Editorials, opinion pieces, quasi-experimental studies, before and after studies, cohort studies (with control), and case-controlled studies were excluded. Observational studies were excluded because known confounders may be only partially controlled and unknown confounders are not controlled.

### 2.2. Search Strategy

We searched and undertook new searches in 15 systematic review databases from 2010 to 2017 to identify similar systematic reviews or protocols and identified none (Table 2).

We searched 16 health professional databases for publications in any January 1990 to December 2017 (Table 2). Authors analyzed subject headings to identify the most efficient search string for each database to capture all relevant articles. An initial limited search of PubMed and CINAHL was undertaken, followed by analysis of text words contained in the title, abstract, and the index terms. Then, all the identified keywords and index terms across all included databases, information sources, and health professional student programs were entered. Initial keywords included tobacco cessation, smoking cessation or nicotine cessation, prevention, intervention, addiction, counseling, education programs, curriculum, curricula, training, and health professionals and individual professions were listed. WorldCat was used to access books and book chapters. Reference lists of all included articles selected for critical appraisal were assessed for relevance based on study title and abstract.

The publishers’ websites of Tobacco Control, Nicotine and Tobacco Research, and Preventive Medicine journals were also searched. The dates and search strategy for each published database are in Appendix A. Searches for grey literature from January 1990 to January 2018, in English, included Theses Canada Portal and ProQuest Dissertations and Theses, the Canadian Agency for Drugs and Technologies in Health (CADTH) database, and relevant documents in five websites (Table 2).

### 2.3. Study Selection

All identified citations were loaded into RefWorks 2.0 (ProQuest LLC, Ann Arbor, MI, USA), and duplicates removed. All studies identified in the database, book, and website searches for relevance based on the information in the title, descriptor terms, and abstract for published and grey literature were assessed. Two independent reviewers screened titles and abstracts and assessed full texts, with disagreements resolved by discussion or by consultation with a third reviewer. The study selection process is in Figure 1 for the published literature and in Appendix A for the grey literature. Of 7853 unique articles from the published literature, 7725 were not further assessed because of inappropriate study design or because they were not relevant to the review topic. After reading the full text, 101 studies of the previously screened 128 articles (78.9%) were excluded because they did not meet the inclusion criteria, resulting in 27 published studies for critical appraisal. Reasons for exclusion upon full-text read are in Appendix A. Nearly all (7914/7930 = 99.8%) of the grey literature articles focused on summaries of evidence, practice guidelines, pharmacotherapy and reports of the need for tobacco intervention education and these were excluded from this review. One unpublished study was selected for critical review, making a total of 28 articles for the critical appraisal process. The grey literature was an unproductive source of information for the review objectives. Reasons for exclusion upon full-text read are given in Appendix A.

### 2.4. Methodological Quality

Two independent reviewers critically appraised all eligible quantitative papers selected for retrieval using the Cochrane Collaboration risk-of-bias tool [73,74] and the JBI critical appraisal instrument [71,72]. The critical appraisal results are in Appendix A. Using both tools provided a rigorous assessment of methodological validity of the educational interventions reported in the meta-analyses [75,76,77,78,79,80,81,82,83,84,85,86,87]. We visually examined funnel plots for student counseling performance, student self-efficacy in tobacco cessation counseling, and for patient smoking cessation at six and 12 months. Although we had fewer than the recommended ten studies [74] in each meta-analysis, the funnel plots showed the studies (n’s for all funnel plots were less than five) to be distributed symmetrically around the mean effect size. This observation may indicate a lower chance of publication bias [74]. The assessment of bias in the meta-analysis studies [75,76,77,78,79,80,81,82,83,84,85,86,87] reflected an unclear risk of selection and performance bias related to incomplete or missing information. We interpreted publication bias in the meta-analyses articles as follows: the impact is not trivial and likely modest, but the major finding is still valid [88] p.286,290.

Thirty-eight per cent of studies used a strong method of randomization and were at low risk of bias for randomization; in 62% it was unclear. In 7.5% of RCTs, the researchers did not know the allocation to the intervention or control groups; in 85% it was unclear and in 7.5%, allocation was disclosed. Those delivering the intervention were blind to group identity in 7.5%; in 85% it was unclear and in 7.5% it was known. In 62%, group identity was unknown to the outcome assessors and in 38%, it was unclear. In 54%, there was either minimal attrition, or an intention-to-treat analysis, or a differential attrition analysis demonstrated no differences between the groups on baseline characteristics or characteristics related to the outcome. In 7.5%, attrition bias was unclear and in 39%, the amount of attrition was of concern. There was no selective reporting detected in 100% of studies (Figure 2 and Figure 3).

### 2.5. Data Extraction

Two reviewers independently extracted data to assess the Population, Intervention, Comparison, Outcome(s), and Study Design construct [72]. Two reviewers piloted the data extraction tool prior to use. Disagreements were resolved by discussion or by consultation with a third reviewer. We contacted authors of primary studies to request missing or additional data. Data for interventions, populations, study methods, student learning outcomes of knowledge, self-efficacy and behavioral performance in tobacco use prevention and cessation counseling, and patient smoking cessation were extracted.

### 2.6. Data Synthesis

Quantitative data from 13 RCTs were pooled for statistical meta-analysis using the Cochrane RevMan program [74]. All results were subject to double data entry. We used a random-effects model [74,88,89] to calculate effect sizes expressed as standardized mean differences (for continuous data), odds ratios (for categorical data) and their 95% confidence intervals. The odds ratio (OR) is consistent with a random-effects model [72,88]. The Cochrane Handbook [73] guided our assessment of heterogeneity in the meta-analysis studies. Heterogeneity was first assessed statistically [74,88] using the standard chi squared and I^2^. If I^2^ is near 0, then nearly almost all of the observed variance is spurious, which means there is nothing to explain. If I^2^ is large (above 70%), it is necessary to assess the reasons for the variance [88] p.119. We conducted sensitivity analyses to explore reasons for heterogeneity and reported the results with each analysis. We interpreted the sensitivity analyses as follows: if results remain consistent across the different analyses, the results can be considered robust. If the results differ across analyses, the results may need to be interpreted with caution [90].

## 3. Results

A total of 28 RCTs in English were identified that included 4343 health care students and 3122 patients. Of the studies sampled, 13 RCTs provided data for meta-analysis. Data from the 15 RCTs [91,92,93,94,95,96,97,98,99,100,101,102,103,104,105] where statistical pooling and meta-analysis were not possible will be reported in a subsequent qualitative analysis. The 28 studies represented five countries: The United States of America (21/28, 78.3%); Switzerland (2/28, 8.7%); Australia (3/28, 4.3%); Denmark (1/28, 4.3%); and Germany (1/28, 4.3%). The language of publications other than English in the area of tobacco use cessation intervention education is in Appendix A.

Students attended universities for 86% of studies, and educational location was not clearly identified in 11% of studies [83,91,100]. In one study (3%), the institute of learning included university plus associate and diploma schools of nursing [103]. Student populations included undergraduate medical students [82,92,95,96,97,98,99,102,104], dental students [76,80], dental hygiene students [86,87], and nursing students [83,100,103]. Carpenter et al. [77] studied groups of students. Graduate medical programs included family practice and internal medicine [75,78,79,81,105], pediatric residencies [93,94], and a surgical residency [84]. Strecher et al. [85] studied internal medicine, family practice, and pediatric residents. Thirteen studies (46%) reported one theory or model; 15 studies (54%) applied two or more theories or models to guide the students’ educational intervention (Appendix A). The reviewed studies focused on cessation; no prevention-focused studies were located. The clients were patients encountered by the health professional students and all were adults.

The systematic review findings regarding knowledge could not be meta-analyzed due to the reported variability in the timing and assessment of knowledge and the lack of consistency of the measurement tools. Knowledge outcomes were focused on the ill health effects of smoking and smoking as a risk factor rather than on smoking cessation counseling (Appendix A).

### 3.1. Meta-Analysis Study Characteristics

Among the meta-analysis articles, the most frequently used models were the 5As model [75,76,78,82,83,84,85,86,87], the Trans-Theoretical Model of Change [78,80,83,86,87], and Self-Efficacy theory [79,81,82,87]. Motivational Interviewing [77,80] was also used as a clinical approach to support tobacco cessation interventions. All studies used a combination of educational strategies. Fifteen percent of the studies [78,83] were conducted outside the United States of America. Meta-analysis study characteristics are given in Appendix A.

The intervention length was variable. Ten studies (77%) implemented educational interventions of two to four hours in duration [75,76,77,79,81,82,84,85,86,87]. Two studies (15%) were eight to 12 hours in length [78,80] and one study [83] (8%) was three days. Eleven studies (85%) utilized a combination of lectures, videotapes, rehearsal of content in role-play with peers, or through interaction with SPs [75,76,78,79,80,82,83,84,85,86,87]. Two studies [77,81] (15%) were delivered via the Internet.

Fidelity refers to direct observation of students using a checklist to monitor that students learned to deliver the intervention correctly and completely. Five studies (39%) identified author or researcher direct supervision [76,79,81,84,85]. Trained preceptors or SPs provided direct observation and feedback to students in three (23%) studies [82,83,86]. No faculty supervision was evident in three (23%) studies [77,78,87], and in two studies (15%) it was unclear if faculty supervision had occurred [75,80].

Certainty of the evidence was either moderate or low using the GRADE method [106,107]. We downgraded the evidence to account for unclear documentation of methods of randomization, high risk of attrition bias, allocation concealment and blinding of participants, and for high levels of heterogeneity and imprecision in self-report of smoking cessation. Within educational institutions, it is challenging to randomize and blind students, who may talk and share information, to create comparable groups. If no detail was available to appraise studies, it does not necessarily mean it was inappropriate, it means it was not reported (Appendix A).

### 3.2. Student Tobacco Cessation Counseling Skills

Ten of the studies assessing tobacco cessation counseling skills assigned them to counsel SPs [76,77,78,79,80,81,82,84] or real patients [75,85]. No SP or real patient co-morbidities were reported. Four studies [76,78,82,84] assessed whether students (n = 1255) demonstrated an increase in 5As tobacco counseling behaviors after instruction compared to control. (Figure 4) The combined studies showed an overall increase in 5As counseling skills (SMD 1.03; 95% CI 0.07, 1.98; *p* < 0.00001, I^2^ 94%). In Figure 4, the I^2^ of 94% indicates considerable statistical heterogeneity and for this group of four studies, the overall group effect is significant (*p* = 0.04). We conducted a sensitivity analysis by temporarily “excluding” Cannick [76] (Year 1), Ockene [82] and Steinemann [84] because of high risk of attrition. When Cornuz [78] and Ockene [82] are excluded, the group effect becomes non-significant at *p* = 0.41 and *p* = 0.14, respectively. When Steinemann [84] is excluded, the I^2^ remains high at 95% but the group results remain significant (*p =* 0.03).

We conducted a second sensitivity analysis by temporarily “excluding” Cannick [76] (Year 1), Cornuz [78], Ockene [82], and Steinemann [84] because of notable differences in magnitude of the reported outcome measures. When Cannick (Year 1) was temporarily excluded because of the magnitude of outcome measures and high performance bias, I^2^ remains high at 94% but the group results remain significant (*p* = 0.02). With Cornuz temporarily “excluded”, I^2^ remains high at 91% and the group effect becomes non-significant (*p* = 0.41) With Ockene temporarily “excluded”, I^2^ remains high at 94% and the group effect becomes non-significant (*p* = 0.14). When Steinemann is “excluded”, the I^2^ remains high at 95% but the group results remain significant (*p* = 0.03). These findings should be interpreted with caution because the I^2^ values above 91% in all studies reveal marked heterogeneity.

Two small studies [77,80] assessed whether students (n = 174) showed an increase in counseling skills for tobacco cessation following a motivational interviewing approach after instruction compared to control (Figure 5). The combined studies showed an overall increase in tobacco cessation counseling skills (SMD = 0.90, 95% CI 0.59, 1.21; *p* = 0.68, I^2^ 0%); the overall group effect is significant (*p* < 0.00001). The I^2^ of 0% indicates that heterogeneity might not be important [73].

### 3.3. Student Self-Efficacy Outcomes

Four studies [80,83,86,87] assessed whether students (n = 1031) showed an increase in smoking cessation counseling self-efficacy after instruction compared to control (Figure 6). The combined studies showed an overall increase in smoking cessation counseling self-efficacy (SMD = 0.38, 95% CI −0.11, 0.87; *p* = 0.05, I^2^ 62%). For these four studies, the group effect is non-significant at *p* = 0.13. The I^2^ statistic of 62% may represent considerable heterogeneity [73]. A sensitivity analysis was conducted “excluding” Rapp [83] because of a high risk of attrition bias. The results with Rapp excluded show a small increase in I^2^ from 62% to 74%, and the overall group effect remains non-significant (*p* = 0.45). The results are similar and, therefore, can be considered robust.

### 3.4. Patient Smoking Cessation Outcomes

Three studies [79,81,85] measured cessation by patients (n = 1615) at six months after counseling compared to control. The combined studies demonstrated an increase in cessation (OR 2.02; 95% CI 1.49, 2.74, *p* = 0.76, I^2^ = 0%) and the group effect was significant (*p* < 0.00001) (Figure 7). The I^2^ = 0% and *p* = 0.76 indicate that heterogeneity may not be important. We conducted a sensitivity analysis temporarily “excluding” Strecher [85] because of high risk of attrition bias. I^2^ remains 0% and the test for overall effect remains strongly positive (*p* < 0.0001). The results are similar and, therefore, can be considered robust.

Three studies [75,78,83] measured cessation (n = 1507) at 12 months after counseling compared to control and the combined studies demonstrated similar cessation rates between the treatment groups (OR 1.04; 95% CI 0.54, 2.01, *p* = 0.04, I^2^ = 69%). The overall group effect was not significant, *p* = 0.91 (Figure 8). Allen’s [75] study assessed smoking status at three months and 12 months after study enrolment. Because of the difference in timing of the first measure (3 compared to 6 months), the Allen study was included with the studies measuring cessation at 12 months. The I^2^ = 69% may represent substantial heterogeneity. We conducted a sensitivity analysis temporarily excluding Rapp [83] and Cornuz [78] because of high risk of attrition bias. The Cornuz study was well executed and if Cornuz is temporarily deleted, the I^2^ decreases to 0% and the group effect remains non-significant (*p* = 0.14). When Rapp is excluded, I^2^ increases from 69% to 82% and the result remains non-significant (*p* = 0.62). Therefore, Cornuz is a source of heterogeneity.

## 4. Discussion

Evidence from our meta-analysis demonstrated that entry-level students can learn brief tobacco counseling skills and can assist patient quitting behaviors. At six months, 78/1000 more patients (than control) counseled by entry-level students reported quitting smoking. These findings are consistent with previously reported research that entry-level education programs are appropriate to enhance health professional student capacity to address client tobacco use [57,62]. Educational institutions can contribute positively to tobacco control measures by implementing entry-level student education programs on treating tobacco use and dependence detailed in Article 12 and 14 of the WHO FCTC [29,108,109]. Brief cessation counseling is politically feasible [110] and cost-effective clinically compared to other medical and disease prevention interventions, and is also a recognized avenue to strengthen health systems globally [30,34,45,46,109].

The majority of entry-level students represented in this systematic review are similar to clinical disciplines offering tobacco dependence counseling: medicine [51,52,53,54,60,61,111,112,113,114,115,116,117,118,119,120,121]; nursing [47,59,62,122,123,124,125,126,127]; pharmacy [10,48,55,128,129]; dentistry [67,130,131,132,133,134]; dental hygiene [9,135,136]; chiropractic therapy [137,138,139]; physical therapy [140]; and midwifery [39,40,141]. Pharmacy [55,56,129] and advanced practice nursing students [58] use the Rx for Change program [56] to counsel against tobacco use. Optometry is considering implementing tobacco dependence education [142].

Eleven of 13 meta-analysis studies utilized a combination of lectures, videotapes, rehearsal or role-play with peers, or interaction with SPs. These findings compare similarly to other basic health professional education programs that use a combination of teaching strategies [136,143,144], role playing [113,123,145], SPs [135], and follow-up contact with participants [145]. SPs assessed student performance and provided feedback in 62% of meta-analysis studies. Feedback is important when learning a new skill and enhances behavioral capacity [146]. The timing and duration of the educational interventions were designed to align student learning with future clinical practice settings, which is important in basic health professional education [55,130]. In addition to multi-component strategies, the most common theoretical approaches were the 5As model, the trans-theoretical model of change, and the motivational interviewing approach to tobacco cessation interventions.

An important consideration is to understand how the effectiveness of student performance compares to that of professionals. Two meta-analysis studies demonstrated that training healthcare professionals to provide smoking cessation interventions had a measureable effect on professional performance. Fiore et al. [30] demonstrated the efficacy of non-physician clinician delivered smoking cessation interventions. Carson et al. [31] found a statistically significant effect in favor of the intervention indicating that trained health care professionals were more likely in all areas to perform tasks of smoking cessation than untrained controls. Our review findings support the premise by Fiore et al. [30] that when health professionals receive training, they are more likely to integrate interventions; when health professionals address their clients’ tobacco use, the odds of successful client quitting increase.

Six studies in our meta-analysis demonstrated a statistically significant intervention effect in student counseling behaviors after instruction compared to control. The strength of evidence was moderate to low. The average change or difference in 5As tobacco-cessation counseling skills with SPs or real patients was significantly higher in the intervention group based on data from 1255 students in four studies. Specific 5As use was reported in three studies [78,82,84]; in one study, students used all of the 5As but were least likely to use Arrange [76]. Health professional student use of the 5As is similar to licensed providers who are asking, assessing, and advising while not assisting or arranging as frequently [147,148,149,150,151]. The average change or difference in tobacco-cessation counseling skills with SPs following a motivational interviewing approach was significantly higher in the intervention group based on data from 174 students in two studies. The models guiding the tobacco-dependence intervention education and the multi-modal learning strategies may have contributed to health professional students’ application of cessation counseling skills.

Student tobacco cessation counseling self-efficacy improved based on data from 1031 students in four studies. The strength of evidence was low. Similar to our findings, Ye et al.’s [152] narrative synthesis regarding pre-licensure students described an increase in confidence or self-efficacy in tobacco cessation counseling outcomes in six of 15 studies. Our review finding, although not statistically significant, is consistent with Social Cognitive theory [153,154,155,156] and is important in skill development [157].

Three of six studies measured cessation by patients at six months after counseling compared to control (OR 2.02; 95% CI 1.49, 2.74, *p* = 0.76, I^2^ = 0%), and the overall group effect was significant (*p* < 0.0001). The strength of evidence was moderate. With a usual curriculum and no education on tobacco dependence cessation counseling, 93 out of 1000 patients will report quitting. With entry-level tobacco dependence counseling, 172 out of 1000 will report smoking cessation, based on 1615 patients in three studies. This finding means that 78 more patients per thousand counseled by entry-level students (than control) will report quitting smoking at six months.

Three studies measured cessation at 12 months after counseling compared to control (OR 1.04; 95% CI 0.54, 2.01, *p* = 0.04, I^2^ = 69%). The overall intervention effect at 12 months was modest and not significant (*p* = 0.91), and the overall strength of evidence was low. With the usual curriculum or no educational program on tobacco dependence counseling, 101 out of 1000 patients will report quitting. The odds ratio at 12 months indicates that with tobacco dependence counseling, only four more patients per 1000 will report smoking cessation, based on 1615 patients in three studies. It was unclear whether students were informed of the ultimate cessation or relapse rates of the patients they counseled. Provision of such feedback may further enhance students’ behavioral capacity. There is an opportunity for future reviews with larger sample sizes to examine the 12-month impact of entry-level student tobacco cessation counseling on patient quitting behaviors.

Tobacco use may account for millions of deaths by 2030 with over 80% of smokers who live in low- and middle-income countries [2,108]. Tobacco-dependence intervention education evidence to date is from high-income countries. The 28 RCTs in this systematic review were from Australia, Denmark, Germany, Switzerland and the United States of America. Guidelines containing the 5As [30] and the Rx for Change [56] models are available at no cost on the Internet. These and other WHO materials [29] may offer low- and middle-income countries available resources for faculty development, student and health care provider education, and policy development. Full implementation of the 5As model, and, in particular, Arrange, may be affected because of varying availability of resources such as inconsistent or no access to Smokers’ Quit Lines.

This systematic review has strengths and limitations. It is the first to focus exclusively on entry-level students. We selected only RCTs because of their ability to control unknown confounders. The theoretical direction and evidence base to guide student instruction and evaluation was evident. Direct observations of student counseling performance contributed to high fidelity of the educational interventions. Although we searched all languages, all included studies in this review were in English and this may represent a bias towards research in English-language countries. Our study may also be at risk of selective outcome reporting bias because only one research protocol was available to critique planned versus reported outcomes, and we did not specify the timing of smoking cessation outcomes ahead of time. Reporting quality versus methodological quality was an issue.

Researchers conducting studies of teaching health professional students’ tobacco dependence counseling should describe in detail the measures they took to reduce risks of bias. Researchers should provide pre- and post-intervention means and standard deviations for control and intervention groups to facilitate future data aggregation. Instruments with strong validity and reliability could reduce heterogeneity arising from variation in outcome measurements. Self-reports of smoking cessation should be validated biochemically to address potential patient deception. Researchers should post research protocols at study outset (e.g., on PROSPERO) to allow for assessment of reporting bias in published studies.

## 5. Conclusions

Academic institutions can play a vital role in tobacco control strategies through tobacco-dependence intervention education. We strongly recommend that faculty in health professional programs based in universities or colleges in all countries begin, or continue, to incorporate tobacco prevention and cessation intervention education into their curricula. Cessation counseling offers patients who smoke a meaningful and safe intervention that considers their preferences. We encourage the use of the 5As model or a motivational interviewing approach with multi-modal learning activities and performance feedback to promote student behavior change. Findings demonstrate that entry-level students can implement brief tobacco cessation counseling and can effect healthier patient outcomes. As future licensed professionals, graduates will have the knowledge and abilities to offer clients brief tobacco cessation counseling tailored to their specific health concerns and address an identified global health priority.

In summary, we recommend:Comparative studies of the 5As, Trans-theoretical Model of Change, Self-Efficacy theory and Motivational Interviewing approaches in appropriately powered and carefully monitored RCTs to assess which approach produces the best results for trainees and patients. Currently, educators use a combination of these approaches.Identify the ideal length of instruction time to optimise outcomes. Currently, it ranges from 8 to 12 hours.Identify the best combination of lectures, videotaping and role play with peers and trained professionals to optimise outcomes.Observe and train preceptors to ensure that they provide optimum feedback and monitor improvements in trainee competence.Observe trainees to ensure they conduct the interventions correctly and completely according to the manual.Measure the counselling skills of trainees and identify those who need more training and provide it.Have trainees observe trained professionals, observe how the trainees implement new skills they observed and discuss how well they implemented them.Ask preceptors to provide feedback on the patient-centredness and quality of the patient–trainee relationship and how to improve it.Maintain good contact with the clients to avoid attrition by the 6- or 12-month follow ups.

## Figures and Tables

**Figure 1 ijerph-16-04158-f001:**
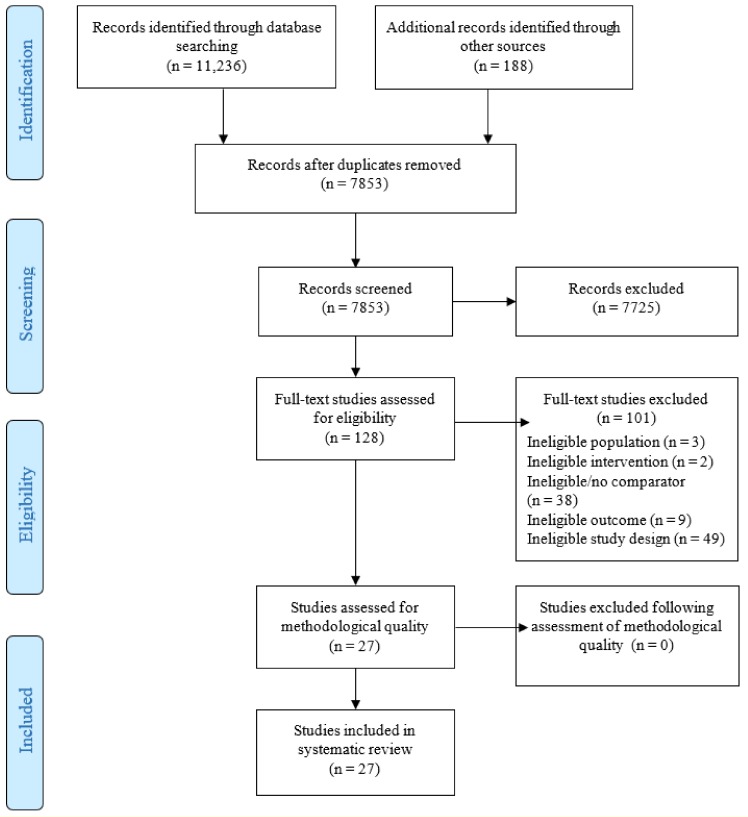
PRISMA flow diagram of search results and study selection process for published studies, 1990–2017. Note: PRISMA = Preferred Reporting Items for Systematic Reviews and Meta-Analyses.

**Figure 2 ijerph-16-04158-f002:**
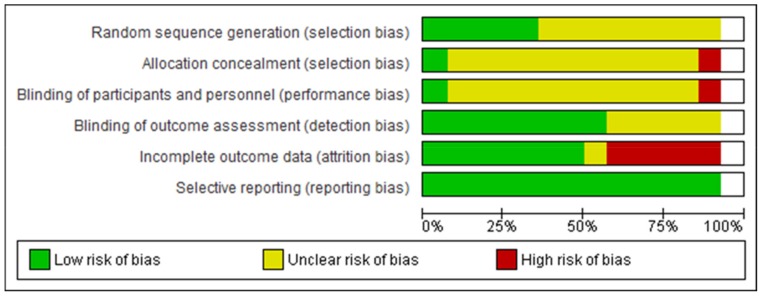
Review authors’ judgements about each risk of bias item presented as percentages across all meta-analysis articles.

**Figure 3 ijerph-16-04158-f003:**
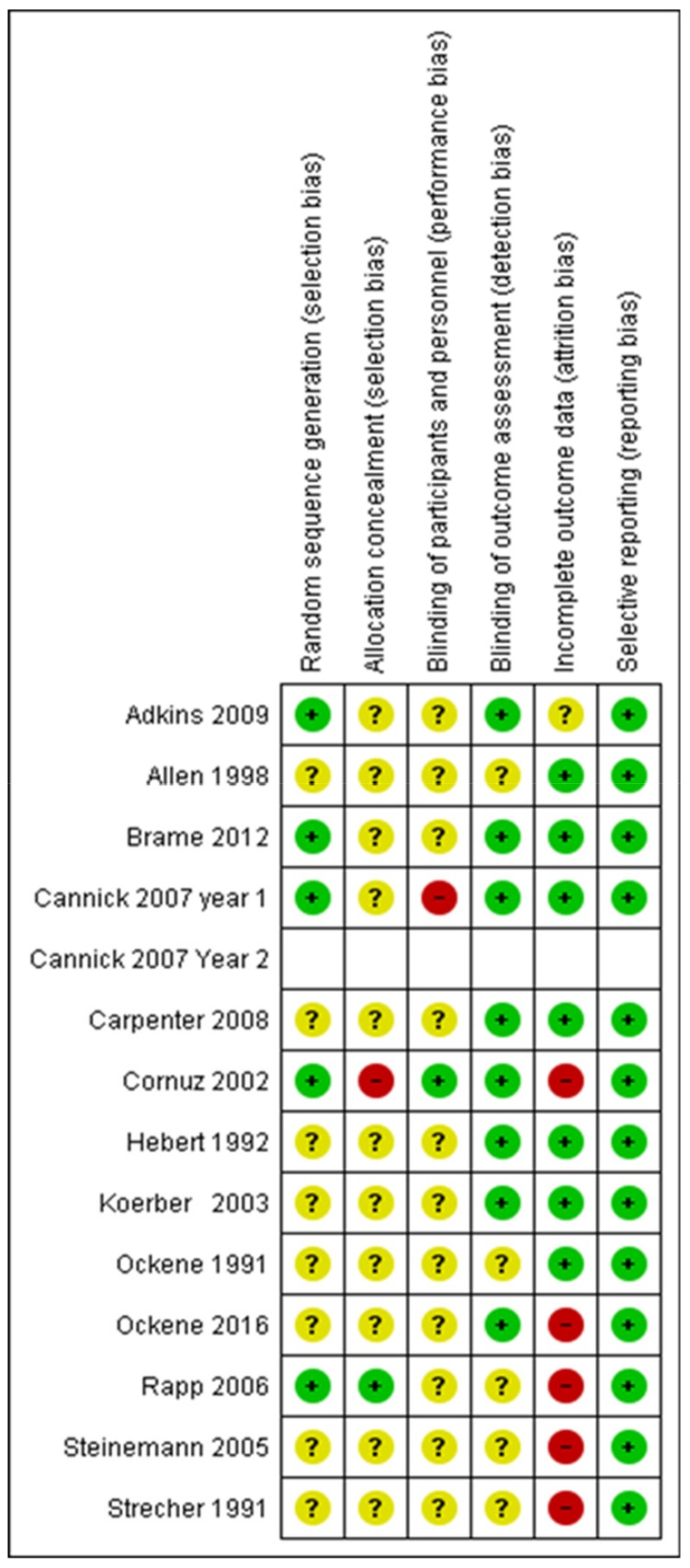
Risk-of-bias summary for meta-analysis articles. Note: To avoid double-counting the risk-of-bias in the assessment for Cannick et al. [76], Year 2 students is left blank. The risk of bias is a summary for both years.

**Figure 4 ijerph-16-04158-f004:**
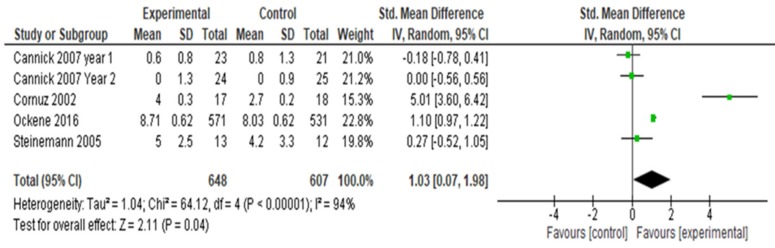
Forest plot for change in tobacco cessation counseling skills using the 5As with standardized or real patients.

**Figure 5 ijerph-16-04158-f005:**
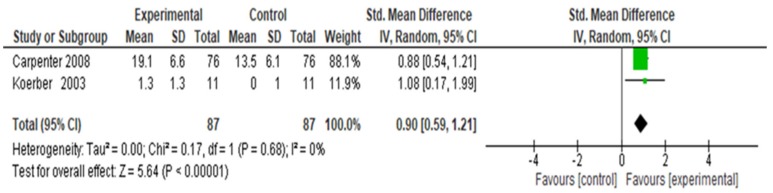
Forest plot for change in tobacco cessation counseling skills with standardized patients guided by Motivational Interviewing.

**Figure 6 ijerph-16-04158-f006:**
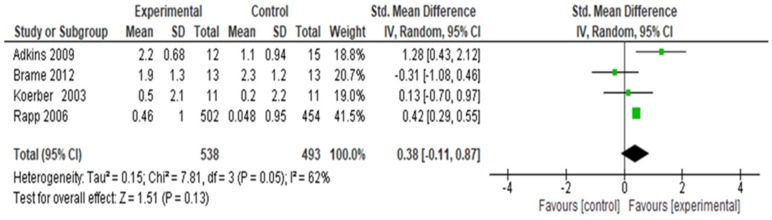
Forest plot for change in students’ self-efficacy in tobacco cessation counseling.

**Figure 7 ijerph-16-04158-f007:**
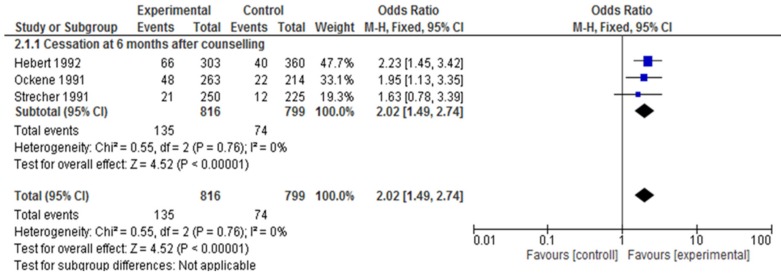
Forest plot for patient smoking cessation at six months after counseling.

**Figure 8 ijerph-16-04158-f008:**
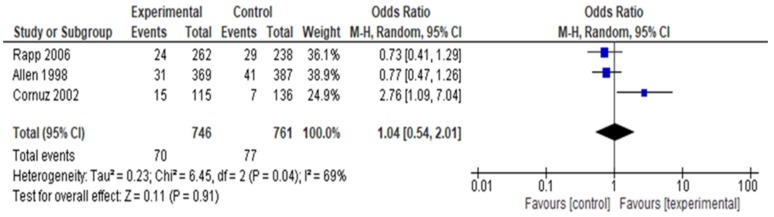
Forest plot for patient smoking cessation at one year after counseling.

**Table 1 ijerph-16-04158-t001:** Health professional program participants.

Health Professional Programs	Post-Graduate Medical Programs
Chiropractic Therapy; Dental Hygiene; Dental Therapy; Dentistry; Medicine; Midwifery; Naturopathic Medicine; Nursing (Registered Nurses, Registered Psychiatric Nurses, Advanced Practice Nurses/Nurse Practitioners; Licensed Practical Nurses); Occupational Therapy; Optometry; Pharmacy; Physical Therapy; Psychology; Respiratory Therapy; Social Work; and Speech Language Therapy	Anesthesia; Cardiac Surgery; Cardiology; Community Medicine; Critical Care Medicine; Dermatology; Endocrinology and Metabolism; Emergency Medicine; Family Medicine; General Medicine; Medical Oncology; Nephrology; Neurology; Obstetrics/Gynecology; Ophthalmology; Orthopedic Surgery; Otolaryngology; Pediatrics; Plastic Surgery; Psychiatry; Radiation Oncology; Respiratory Medicine; Surgery; and Vascular Surgery

**Table 2 ijerph-16-04158-t002:** Databases searches.

Systematic Review Databases	Published Studies Databases	Grey Literature
JBI Database of Systematic Reviews and Implementation Reports; the Cochrane Library of Systematic Reviews (including the Tobacco Addiction Group Reviews); the Campbell Collaboration Library. The National Health Centre Reviews and Dissemination databases [Database of Systematic Reviews; Health Technology Assessment; Economic Evaluation Database]; Health Technology Assessment International; Evidence of Policy and Practice Information; Physiotherapy Evidence Database; Occupational Therapy Systematic Evaluation of Evidence; PROSPERO; PubMed; CINAHL and Epistemonikos from July 2015	Allied and Complementary Medicine; CINAHL; PubMed; EMBASE; Scopus; SocIndex; PsychInfo; Academic Search Premier; Education Resources Information Center; Education Search Complete; Health Source-Nursing/Academic Edition; Translating Research into Practice; Google Scholar; Web of Science; and Natural Standard	Theses Canada Portal; ProQuest Dissertations and Theses; CADTH; Directory of Grey Literature via New York Academy of Medicine website; websites of the Canadian Council of Tobacco Control (until March 30, 2012); Canadian Centre on Substance Abuse; Health Canada; and Canadian Public Health Association

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
