# Peer review of "The Effectiveness of Tobacco Dependence Education in Health Professional Students’ Practice: A Systematic Review and Meta-Analysis of Randomized Controlled Trials"

_ijerph, 2019, doi:10.3390/ijerph16214158_

Round 1
Reviewer 1 Report
General:
This is a nice review of student-level education of tobacco cessation treatment. The premise is a good one and one that I imagine many training programs will be able to use to justify including tobacco treatment in their curricula. With that in mind, I think the paper would benefit from a slight change in focus from justifying the review (it’s nicely done and informative) to demonstrating the results of how useful it is for patient care to train students to treat smokers. I will leave this to the editor to decide if such a shift would be of interest to their readers. However, this is my preference and I will provide some comments as to why below. In addition, a few general comments:
The authors use several different terms related to tobacco use and it is important that they are consistent and relate directly to the statistics that are cited throughout the paper. Are they referring to all tobacco products or just cigarettes alone? There is “tobacco dependence education” in the abstract (To me, this means teaching students about addiction in general and how nicotine dependence and withdrawal fit this model). And there is a statement of what tobacco dependence treatment includes, but here it’s just cigarette smoking (introduction, line 50). Throughout the introduction, however, there is also “Tobacco use”, “tobacco smoking”, “cigarette smoking”. Along a similar line, the authors discuss tobacco dependence education as their primary question. Were the studies they reviewed inclusive of all tobacco products? Or were they specific to cigarettes? Or was this unclear? This should be included in the If someone read this article and decided they should include “tobacco treatment” in their health professional training, this would be a useful detail to consider. In the introduction and abstract, the authors discuss different theoretical underpinnings for the education that students received. It reads to me as if these things were considered as mutually exclusive, which of course they are not. I see in the supplementary files that the authors did not consider these things to be exclusive either. This leads me to wonder how they did stats where the different theories were compared. However, to me this distinction isn’t really the point and I don’t know that this is a detail that matters. More importantly, discussing these as if they are mutually exclusive leads the reader, if they were new to tobacco treatment, to think that one may be better than another (when in fact they are often all used in concert). The main point of the paper is to demonstrate that teaching students to provide tobacco treatment is important and useful. If you were to consider changing the focus of the manuscript, supplementary table #2 and 3 do a nice job of demonstrating that there are different ways to go about doing the treatment and assessing knowledge for students. I think these are much more interesting tables than some of the tables in the paper (Table 1 or 2, and figure 2 and 3).
Abstract:
there seems to be a typo starting on line 29 where there is a period, but the sentence continues.
Introduction:
When providing a summary of the impact of smoking on health, I think the most problematic issues should be discussed first so that the reader has a sense of what the major problems are. In this case, it would be CVD (heart attack, stroke), COPD and cancer. 2.1.4-the introduction suggests that you will include smoking “behaviors” but the only behavior is quitting (rather than measures of treatment compliance, medication use, cigarette logs, etc.). Perhaps you could amend the intro to reflect this. Also, there is some suggestion that there will be a discussion of smoking prevention skills. However this does not appear to be included at all in the main body of the paper or the analyses. The three main outcomes are knowledge, self-efficacy, and clinical performance for treating (not preventing) tobacco use. This should be removed. Maybe I missed this but was the treatment education focused on adults only? Or kids too?
Methods:
the sentence starting on line 171 is long and confusing can it be rephrased?
Discussion:
The topic of this analysis is a good one and the results have a relatively clear cut interpretation-- Teaching students of health professions about treating tobacco cessation is a good idea. The discussion would be strengthened if there was less of a focus on re-iterating the results and instead a focus placed on putting the study in the context of what has been done before in terms of teaching health care professionals. If there are no other reviews of students then what about more on existing professionals and their training. How does student training differ, for instance? As mentioned above, I also think this paper would be very useful for others who are considering, or running, education for their students. Why not be more overt in providing helpful suggestions for what they may want to do in their programs. You clearly have insight here that others would appreciate and could use.
Reviewer 2 Report
This is an excellent paper describing a piece of work that addresses a hitherto unresolved question, that of the evidence in support of training entry-level health care professionals.
The topic aligns with the interests of the journal.
I have very few criticisms to make because it is quite simply one of the best articles I have come across in terms of writing, description of rationale, methods, findings and interpretation. There is an extensive set of references. The article is virtually flawless - I could not spot a typographical error anywhere.
I think the abstract could be improved by including a mention in the abstract that this is a systematic review (it is appropriately noted in the title but not in the abstract, which it should be if it had been written as a structured abstract following the IMRAD format) and including the date when it was done.
Figures 1 and 3 both have titles with additional explanatory information but this information should be presented as footnotes below the titles.
But these are small matters. The authors describe a meticulously conducted systematic review that draws on all the required and relevant benchmarks/standards/guidelines and present the findings of the meta-analysis thoroughly and appropriately. I have no concerns about the search strategy, selection methods and criteria, nor of the statistical approach to the meta-analyses - all align with the methods in the Cochrane Handbook.
The results are presented clearly in text and figures.
The conclusions drawn are appropriate given the findings, and are important and need to be promoted.
Are the authors planning to include their work as a Cochrane review to complement the current (now quite dated) 2012 review of training health professionals?
